# Role of Serum Free Light Chain Assay in Relapsed/Refractory Multiple Myeloma. A Real-Life Unicentric Retrospective Study

**DOI:** 10.3390/cancers13236017

**Published:** 2021-11-29

**Authors:** Uros Markovic, Alessandra Romano, Claudia Bellofiore, Annalisa Condorelli, Bruno Garibaldi, Anna Bulla, Andrea Duminuco, Vittorio Del Fabro, Francesco Di Raimondo, Concetta Conticello

**Affiliations:** 1Division of Hematology, Azienda Policlinico-OVE, University of Catania, 95124 Catania, Italy; alessandra.romano@unict.it (A.R.); claudia.bellofiore@studium.unict.it (C.B.); uni325265@studium.unict.it (A.C.); uni334269@studium.unict.it (B.G.); anna.bulla24@gmail.com (A.B.); a.duminuco@studium.unict.it (A.D.); vdelfabro@yahoo.it (V.D.F.); diraimon@unict.it (F.D.R.); ettaconticello@gmail.com (C.C.); 2Oncohematology and BMT Unit, Mediterranean Institute of Oncology, 95029 Viagrande, Italy; 3Postgraduate School of Hematology, University of Catania, 95124 Catania, Italy

**Keywords:** relapsed/refractory multiple myeloma 1, serum free light chains 2, predictive factor 3, relapse type 4

## Abstract

**Simple Summary:**

According to the International Myeloma Working Group diagnostic criteria serum free light chain (sFLC) assay is recommended for disease monitoring in oligo-secretory and micromolecular multiple myeloma (MM). However, normo-secretory patients could suffer from oligo-secretory/micromolecular escape at disease relapse, making it difficult to monitor disease relapse with serum protein electrophoresis alone. The possibility of a simple and manageable biochemical exam able to predict either future treatment response at disease relapse or relapse type in the course of treatment could be of great aid. In this work, we retrospectively analyzed the occurrence of oligo-secretory/micromolecular escape at disease relapses and its impact on disease outcome. Moreover, we have retrospectively evaluated the role of sFLC ratio and involved sFLC at disease relapse in a historical real-life single-center cohort of MM patients that underwent at least three lines of treatment.

**Abstract:**

Background: In the era of novel drugs a growing number of multiple myeloma (MM) patients are treated until disease progression. Serum free light chain (sFLC) assay is recommended for disease monitoring in oligo-secretory and micromolecular MM. Methods: In this real-life survey, a total of 130 relapsed/refractory MM patients treated at our center with at least three lines were investigated as a retrospective cohort. Results: The median age at diagnosis was 64 years and more than half of patients were male. A total of 24 patients (18%) had oligo-secretory/micromolecular disease at diagnosis. More than 20% of 106 normo-secretory patients had oligo-secretory/micromolecular escape. In order to evaluate potential role of sFLC assay before (“pre”) and after (“post”) every treatment line, involved serum free light chain values (iFLC) less than 138 mg/mL and serum free light chain ratios (FLCr) <25 were identified by using ROC curve analysis. The analysis of the entire cohort throughout four treatment lines demonstrated a statistically significant negative impact on progression-free survival (PFS) for both involved pre-sFLC and its ratio (respectively *p* = 0.0086 and *p* = 0.0065). Furthermore, both post-iFLC and post-FLCr greater than the pre-established values had a negative impact on PFS of the study cohort; respectively, *p* = 0.014 and *p* = 0.0079. Odds ratio analysis evidenced that patients with both involved post-sFLC greater than 138 mg/mL and post-FLCr above 25 at disease relapse had a higher probability of having clinical relapse (respectively *p* = 0.026 and *p* = 0.006). Conclusions: Alterations of sFLC values, namely iFLC and FLCr, both prior to treatment initiation and in the course of therapy at every treatment line, could be of aid in relapse evaluation and treatment outcome. We therefore suggest close periodical monitoring of sFLC assay, independently from secretory status.

## 1. Introduction

Multiple myeloma (MM) is characterized by the clonal proliferation of malignant plasma cells in the bone marrow and the presence of paraprotein in serum and urine, associated with organ dysfunction [1]. The serum paraprotein can be represented by an intact immunoglobulin monoclonal protein (M-protein), consisting of heavy chains linked to light chains, or solely, by the serum free light chains (sFLC), kappa or lambda, detectable through immunofixation electrophoresis and immunohistochemistry, respectively [2]. The International Myeloma Working Group (IMWG) set the thresholds of 1 g/dL serum M-protein and 200 mg/24 h urine M-protein to define a measurable disease, while lower levels of M-protein identify unmeasurable disease, which is further classified as oligo-secretory (OS), non-secretory (NS) or non-producer MM [3]. Oligo-secretory MM is characterized by low levels of M-protein in serum or urine, while non-secretory MM, also called micromolecular light chain MM, is characterized by positive monoclonal kappa or lambda light chains detected by immunohistochemistry and the absence of M-protein in immunofixation. Non-producer MM, on the other hand, has neither detectable kappa or lambda light chains by immunohistochemistry nor positive immunofixation.

In MM, there is an overproduction of only one type of monoclonal FLC (involved FLC, iFLC) leading to an imbalance between kappa and lambda sFLC and an increase of involved/uninvolved sFLC ratio (FLCr), that represents a quantitative marker of increased monoclonal (involved) light chain and the decrease of the polyclonal (uninvolved) one. Since sFLC have a considerably shorter lifespan than M-protein—with half-lives around 21 days—they can reveal early response to treatment. Thus, sFLCs represent a pivotal tool for monitoring MM patients, especially in those affected by NSMM in which M-protein is undetectable. Indeed, sFLC evaluation has already been recommended and is currently part of the by IMWG guidelines for MM diagnosis and response assessment [4]. However, the possible switch from measurable to unmeasurable disease—the so-called “oligo-secretory/micromolecular escape” described already by Larson and colleagues—makes it difficult to diagnose disease relapse in this patient setting, according to the current response assessment guidelines [5].

There are different methods of measuring FLCs: some are based on the use of monoclonal antibodies, such as N-Latex Siemens and Seralite lateral flow assay, while others are polyclonal antibody-based assays, such as the Freelite. Several studies have compared the available assays showing different results in measuring FLCs, hence it is recommended to use only one assay to monitor patients [6,7]. In the updated IMWG diagnostic criteria the only recognized sFLC assay was Freelite [8]. Serum FLC ratio has a prognostic significance both at diagnosis, with a ratio greater than 100 being part of the myeloma-defining events (MDE), and at response evaluation with the presence of normal FLCr in a complete, stringent response [9]. Despite this relevant role at diagnosis and for the evaluation of response, its role as a prognostic element during disease course and at every relapse needs further study to be better assessed.

In this work, we retrospectively analyzed the occurrence of oligo-secretory/micromolecular escape at disease relapses and its impact on disease outcome. Moreover, we have retrospectively evaluated the role of sFLC ratio and involved sFLC at disease relapse in a historical real-life single-center cohort of MM patients that underwent at least three lines of treatment.

## 2. Materials and Methods

### 2.1. Patient Selection

In this real-life survey a total of 130 relapsed/refractory multiple myeloma (RRMM) patients treated at our center between January 2000 and October 2020 were investigated as a retrospective cohort. Data was collected from the medical records of the patients treated with at least three lines of therapy, including novel agents. The study was approved by an independent ethics committee of the coordinating center (*Policlinico Catania 1*, n.34/2019/PO) and was conducted in accordance with International Conference on Harmonization Guidelines on Good Clinical Practice and the principles of the Declaration of Helsinki.

Patients included were evaluated at diagnosis and at every disease relapse, according to Eastern cooperative oncology group (ECOG) scale of performance status (PS). The cohort was also classified according to age, sex, time of diagnosis, type of monoclonal component [immunoglobulin G (IgG), IgA and micromolecular, kappa or lambda], secretory status (measurable intact immunoglobulins, oligo-secretory and light-chain/micromolecular) and number of treatment lines. Secretory status was classified in measurable and unmeasurable disease (divided into oligo-secretory and micromolecular light chains), according to IMWG guidelines [10,11]. The secretory type was followed after every relapse, using the eventual change from measurable to non-measurable disease to measure both the M-protein and sFLC assay.

### 2.2. Disease Characteristics at Diagnosis and Relapse

Characteristics of myeloma disease, such as fluorescent in-situ hybridization (FISH) analysis and elevated LDH level (above upper normal limit) were evaluated at diagnosis and after every relapse. The Revised International Staging System (R-ISS) was evaluated only at diagnosis. FISH analysis was considered as high risk in the presence of the following alterations present in at least 10% of purified plasma cells: t(4;14), t(14;16); del 17p13; del 1p32; gain 1q21 together with another cytogenetic alteration. All other cytogenetic alterations were considered as standard risk [12]. Staging was evaluated as standard risk, stage I-II according to R-ISS (B2 microglobulin <5.5 mg/dL) and high risk, stage III with B2 microglobulin ≥5.5 mg/dL, together with elevated LDH level or high-risk cytogenetics [13].

Myeloma patients undergoing treatment were monitored with monthly protein electrophoresis for M protein measurement, serum immunofixation (in case of absence of M protein) and monthly sFLC assay when available. Serum FLC was assessed with the Siemens N latex^®^ assay, according to the center’s availability. Involved sFLC (iFLC) and involved/uninvolved FLCr were evaluated according to IMWG guidelines [11]. The sFLC values were defined as “pre-sFLC” when evaluated before treatment initiation (e.g., sFLC assay prior to first treatment line: 1st line pre-sFLC; sFLC prior to second line: 2nd line pre-sFLC, etc.), and “post-sFLC” at disease relapse following last therapy (e.g., sFLC assay at first disease relapse: 1st line post-sFLC, sFLC at second disease relapse: 2nd line post-sFLC, etc.).

Response evaluation and progression assessments were reported according to the International Myeloma Working Group consensus criteria for measurable and unmeasurable disease [4,14]. In our institution, whole-body low-dose computed tomography (WBLDCT) and magnetic resonance imaging (MRI) of the spine and pelvis are used as routine imaging techniques for diagnosis and progression assessment, while positron emission tomography (PET) scan is generally used in case of possible extramedullary disease (EMD) progression. The sFLC assay was included in disease evaluation when available, independently of secretory status at diagnosis. Oligo-secretory/micromolecular escape was defined as the disease relapse of a normo-secretory MM patient in the presence of M-protein <1 g/dL, confirmed at subsequent treatment lines. Disease relapse was evaluated after every relapse and classified as biochemical or clinical according to IMWG criteria [15].

Given the retrospective cohort design with important heterogeneity in disease management between patients caused by drug availability, treatment combinations were evaluated on the basis of their duration (continuous versus fixed duration treatment) and number of novel drugs used (doublet versus triplets). Lenalidomide refractoriness was also evaluated at every disease relapse.

### 2.3. Statistical Analysis

Descriptive statistics were generated for data analysis. Qualitative results were summarized by counts and percentages. Descriptive analysis was performed by frequency distribution for continuous variables. Pearson’s chi-squared test was used to determine the absence of statistically significant difference between the expected and the observed frequencies in the examined categories. A receiver operating characteristic (ROC) curve was used for all patients with available free light chain values at disease relapse throughout the four treatment lines, in order to analyze and determine optimal threshold values of iFLC and sFLC ratio. Pre-sFLC values were compared to the progression-free survival (PFS) status of each patients’ treatment line in order to predict populations with poorer outcome. On the other hand, post-sFLC values were compared with relapse type (biochemical versus clinical), which could predict patients with higher probability of clinical relapse according to sFLC values.

Survival analysis of the general population was then estimated with the Kaplan−Meier method in terms of progression free survival (PFS), based on the sFLC threshold values from ROC analysis, and compared by the log-rank test in the first four treatment lines (PFS1 to PFS4) and for the entire cohort throughout all four treatment lines (Cohort’s PFS). PFS was calculated from the time of each treatment start until the date of progression, relapse, death or date the patient was last known to be in remission. Accordingly, PFS1 was defined as time from the start of therapy until first relapse, PFS2 as from the start of the second line until second relapse and PFS3 as from the start of the third line until third relapse and so on. Overall survival (OS) was calculated from the start of first treatment line until the date of death for any cause or the date patient was last known to be alive.

Pre-sFLC assay was used to estimate the predictive significance of baseline values prior to treatment start on subsequent PFS. On the other hand, high post-sFLC values were evaluated as an additional diagnostic tool for disease relapse, following last therapy, in terms of PFS. As for the impact of post-sFLC values on relapse type (biochemical versus clinical) odds ratio (OR) was used. A two-tailed *p* value < 0.05 was considered to be statistically significant, with 95% confidence intervals (95% CIs) for both Kaplan–Meier and OR analyses.

All calculations were performed using MedCalc version 12.30.0.0 (Producer: MedCalc Software bvba, Ostend (Belgium), www.medcalc.org, accessed on 10 December 2020.

## 3. Results

### 3.1. Patient’s Characteristics at Diagnosis

One hundred and thirty MM patients were followed from diagnosis until the date of last follow-up; median age at diagnosis was 64 years (range 31–80 years), while median age at last follow-up was 71 years (range 37–88 years), with more than half of patients being male. About 80% of evaluated MM patients at diagnosis had a standard risk of cytogenetic alterations, while the rest were at high risk. Eight out of twenty-eight patients with available sFLC values at diagnosis (29%) had FLCr greater than 100, as a myeloma-defining event, along with the presence of least one CRAB criteria prior to treatment start. Clinical characteristics of the patients at diagnosis, including age, sex and disease characteristics (type of monoclonal component, time of diagnosis, etc.) are described in Table 1. As for aggressive disease presentation, a total of four patients had extramedullary disease at diagnosis, while, in one patient, plasma cell leukemia was diagnosed.

Median follow-up was 82 months (range 6–226 months); OS for the whole population was 133 months (CI 95% 113–149 months), with 55 patients (42%) alive at last follow-up. A patient’s flow diagram, kept throughout the first four treatment lines, is described in Figure 1.

### 3.2. Secretory Status, Treatment Response and Relapse Type

Treatment characteristics, such as number of lines of treatment, autologous or allogeneic transplantation, and secretory status are described in Table 2. All patients had secretory disease. The median number of treatment lines was four (range 3–8); 45% of patients were exposed to more than four lines of treatment. A total of 63 patients (48%) underwent at least one ASCT, and ten patients underwent both autologous and allogeneic stem cell transplantation (16%). Fourteen out of 106 (12%) normo-secretory patients at diagnosis switched to oligo-secretory status at first disease relapse. The escape was confirmed in another four patients at second relapse, and at last follow-up a total of twenty-one normo-secretory patients at diagnosis (28%) underwent oligo-secretory switch. Furthermore, micromolecular and non-producer switch occurred after the first two lines of treatment in four patients each, respectively. The median M-protein value in oligo-secretory patients was 0.15 g/dL (range 0.1–0.95 g/dL).

Novel agents were used in more than 90% of treatment regimens, including proteasome inhibitors, mainly bortezomib and carfilzomib, and immunomodulatory drugs, such as thalidomide, lenalidomide and pomalidomide. Additionally, daratumumab and, less frequently, elotuzumab were utilized in more than half of the cohort from the third line onwards. On the other hand, chemotherapy regimens were more frequent in the advanced treatment lines, mostly for the debulking of heavy tumor burden as a bridge to the next treatment. Given the heterogeneous drug regimen combinations over the study period of 20 years, therapies were classified based on treatment duration, namely continuous versus fixed duration, and are described in Table 3. The most frequently used fixed duration regimens were bortezomib-based, alone or in combination with thalidomide, melphalan, cyclophosphamide, bendamustine or adriamicin. As for continuous regimens, immunomodulatory drugs, such as pomalidomide with dexamethasone and lenalidomide, alone or together with carfilzomib, ixazomib, cyclophosphamide and melphalan were mainly used. Finally, monoclonal antibodies were part of continuous treatments with daratumumab as a single agent, or in combination with lenalidomide or bortezomib and elotuzumab together with lenalidomide. All treatment regimens were associated with corticosteroids, in most cases, dexamethasone. Furthermore, in patients exclusively treated with novel agents with corticosteroids, treatment was classified to doublet and triplet regimens.

Responses were classified in four groups: deep responses such as complete response (CR) and very good partial response (VGPR), partial response (PR) and no response (less than PR). High rates of deep response after the first line of treatment in half of the patients continued to drop; on the contrary, in advanced therapies, PR achievement and lack of response were present more frequently, as expected.

Clinical relapse, based on the presence of CRAB criteria, was present in 70–80% of patients at every disease relapse, compared to biochemical one. Ten patients had extramedullary relapse, along with four patients at diagnosis, while three patients had plasma cell leukemia progression. Lenalidomide refractory status was evaluated at every disease relapse with increasing number of patients in advanced treatment lines (Table 3). Disease characteristics throughout treatment lines, including secretory, cytogenetic status, LDH level and patient performance status, in terms of ECOG, are also described in Table 3.

### 3.3. Serum Free Light Chain Response Evaluation at Diagnosis and Disease Relapse

Serum free light chains and their ratios were estimated based on availability, and the number of evaluated patients, although limited, increased between the first and fourth disease relapse. In order to evaluate predictive role of sFLC assay in a limited population, we performed ROC analysis of the study cohort throughout the four treatment lines based on PFS status, a total of 195 treatment initiations with available pre-sFLC values, evidencing the significance of pre-iFLC values greater than 138 mg/mL (*p* = 0.006) (Figure 2). On the other hand, ROC analysis of 229 post-FLC values of the entire cohort was compared with relapse type (biochemical versus clinical) confirming iFLC value greater than 138 mg/mL and ratio greater than 25 as threshold values, although without statistical significance. Around 60% of evaluated patients had an iFLC level greater than 138 mg/L according to a Siemens N-latex^®^ assay throughout different treatment lines, while around 10% had no sFLC alteration. An FLC ratio greater than 25 was present in about half of the study population at disease relapse between first and fourth treatment line, and greater than 100 in about 30%.

Serum FLC assay was used monthly in 115 patients throughout the four treatment lines, prior to every disease relapse as part of response evaluation and is described in Table 4. Around 60–70% of the patients had altered sFLC values prior to disease relapse, including both clinical or biochemical. The sFLC values were accompanied or succeeded by M-protein increase in more than half of them, while in the rest sFLC assay was the only serological predictor of disease relapse. The first sFLC alteration outside of normal range was mainly observed 6 months prior to confirmed disease relapse in patients with regular monthly sFLC monitoring (Table 4).

### 3.4. Predictive Parameters at Relapse: Role of Serum Free Light Chains

The patients included in this study were evaluated over a 20-year period. PFS at every relapse was evaluated and, as expected, decreased from the previous to the successive line of treatment (Figure 3).

In the attempt to demonstrate the significance of sFLC assay at disease relapse in terms of PFS, involved free light chains and FLCr, both at disease relapse prior to treatment start, “pre-sFLC”, and following last therapy, “post-sFLC”, for the entire cohort and from 1st to 4th treatment line, were evaluated by using univariate Kaplan Meier analysis (Table 5).

The analysis of the entire cohort throughout four treatment lines demonstrated a statistically significant negative impact on PFS for both involved pre-sFLC greater than 138 mg/mL and its ratio ≥25 (respectively *p* = 0.0086 (HR 1.53, 95% CI 1.12–2.11) and *p* = 0.0065 (HR 1.58, 95% CI 1.14–2.19)) (Figure 4A,B). The impact was confirmed in the 3rd treatment line for both pre-iFLC and ratio, respectively *p* = 0.0002 and *p* = 0.0009, and the 4th line involved pre-sFLC (*p* = 0.03).

Furthermore, both post-iFLC and post-FLCr greater than the abovementioned threshold had a negative impact in terms of PFS of the study cohort; respectively, *p* = 0.014 (HR 1.39, 95% CI 1.07–1.81) and *p* = 0.0079 (HR 1.42, 95% CI 1.09–1.86) (Figure 4C,D). Post-FLCr remained significant in both third and fourth treatment lines; respectively, *p* = 0.03 and *p* = 0.01. In order to evaluate the negative effect on PFS, post-iFLC and post-FLCr were analyzed together with relapse type (biochemical versus clinical) of the study cohort—a total of 229 disease relapses—by using odds ratio. Patients with both involved post-sFLC greater than 138 mg/mL and post-FLCr above 25 at disease relapse had a higher probability of having clinical relapse (respectively OR 2.01 (95% CI 1.09–3.79), *p* = 0.026 and OR 2.55 (95% CI 1.31–4.9)], *p* = 0.006). However, multivariate analysis failed to confirm the statistically significant value of both pre-sFLC and post-sFLC values in terms of overall PFS.

Patients were next analyzed separately in different cohorts. The study population was divided based on treatment type to triplet and doublet regimens, that were subsequently analyzed with the abovementioned sFLC values. Both pre-iFLC and pre-FLCr remained significant independently from regimen type, *p* = 0.03 and *p* = 0.004 in the triplets, respectively, and *p* = 0.02 in each doublet regimen. As for the post-sFLC values, the impact of post-iFLC and post-FLCr remained significant in the doublets, respectively, *p* = 0.001 and *p* = 0.005, while their impact in the triplet regimen was not confirmed (*p* > 0.05) (Table 6).

Similarly, treatment type was divided to continuous and fixed-duration based on duration type and subsequently analyzed. Both pre- and post-sFLC values remained significant in the course of continuous treatment; respectively, *p* = 0.0002 and *p* = 0.0001 in pre-iFLC and pre-FLCr and *p* = 0.01, while *p* = 0.0007 for post-iFLC and post-FLCr, respectively. The same result was not confirmed for fixed-duration treatment, where statistical significance was not reached neither for pre- nor for post-sFLC values (Table 6).

Finally, the oligo-secretory/micromolecular subcohort of the entire study population throughout the four treatment lines was analyzed using an ROC curve and iFLC ≥ 378 mg/mL and FLCr ≥ 115 were evidenced as cut-off values. However, univariate analysis failed to confirm statistical significance with the abovementioned sFLC values (*p* > 0.05) in the oligo-secretory/micromolecular cohort. On the other hand, pre-iFLC ≥ 138 mg/mL and pre-FLCr ≥ 25 confirmed statistical significance even in the oligo-secretory/micromolecular cohort, respectively *p* = 0.03 and *p* = 0.009, while post-sFLC values did not have an impact in terms of PFS (Table 6).

A total of 10 patients with extramedullary disease at diagnosis or disease relapse were evaluated with sFLC analysis. Given the extremely limited patient subcohort, serum free light chains did not demonstrate statistically significant impact in terms of PFS, neither for pre- nor post-sFLC values at abovementioned cut-offs (*p* > 0.05).

## 4. Discussion

Over the last 20 years, the well-known prognostic factors used at diagnosis have been refined. These factors allowed us to stratify patient’s prognostic risk and verify the efficacy of upfront anti-MM first-line treatment. However, with increasing number of available therapies in RRMM setting potential prognostic factors at disease relapse in terms of PFS could have a pivotal role for treatment decisions during patients’ journey, not only at their beginnings. Indeed, IMWG guidelines suggest that Binding Site^®^ evaluation of involved and sFLC ratio represents a useful prognostic factor at diagnosis [4], although its impact at disease relapse remains largely unknown.

To this aim, we collected and reviewed data from 130 RRMM patients treated with at least three lines of therapy including novel agents and analyzed them retrospectively. In real-life populations, extremely aggressive clinical relapse can, per se, predict poor outcome, particularly in the presence of extramedullary disease. However, the possibility of a simple and manageable biochemical exam, able to predict or anticipate biochemical or clinical relapses after first remission, could be of great aid. In order to evaluate its utility, the involved serum free light chains and its ratio were monitored, based on their availability.

Even though in our cohort, only 20% of patients was oligo-secretory/micromolecular at diagnosis, around 25% shifted from normo-secretory to oligo-secretory/micromolecular form, mainly in the first three lines of therapy. Larson et al. described in a letter to the editor the concept of “oligo-secretory escape”, characterized by an important increase of patients with unmeasurable disease after treatment start compared with diagnosis, respectively, 49% and 9%. They concluded by recommending the sFLC assay for response assessment [5]. It has also been shown that non-secretory/light-chain escape could have more aggressive disease course with increased rates of EMD in almost one third of the population [16]. Therefore, the use of regular monitoring by sFLC assay should be performed independently of the secretory status and could be of aid in the early detection of oligo-secretory/micromolecular escape. In a study by Kühnemund and colleagues, the shift to oligo-secretory, micromolecular or non-producer disease could be interpreted as a possible sign of disease de-differentiation and transformation to EMD [17]. As widely described and confirmed in our cohort, the duration of treatment and length of the treatment-free interval progressively decreases with advanced treatment lines [18].

In our study cohort, sFLC assay was altered in 60–70% of evaluated patients at relapse. It was mostly followed or accompanied by M-protein increase. Given that, in most of the patients, the alteration was present at least several months before the relapse, the interpretation of the sFLC results alone or together with M-protein could be of aid for the early diagnosis of biochemical relapse. According to IMWG, biochemical relapse should be treated in high-risk patients—i.e., those with aggressive disease at diagnosis or relapse or adverse cytogenetics and short treatment-free interval, or with a suboptimal response—in order to avoid severe symptomatic disease [19,20]. In a previous work, our group has already suggested extending the application of sFLC to first and subsequent relapse, since it had an anticipatory role in clinical relapse, including cases of aggressive non-secretory or oligo-secretory EMD disease [21]. This suggestion comes from four clinical cases in which sFLC evaluation preceded clinical aggressive relapse.

In this work, the importance of sFLC assay at relapse was investigated retrospectively, both prior to treatment start (pre-sFLC) and at disease relapse after last therapy (post-sFLC). Involved sFLC value greater than 138 mg/mL value and ratio of 25 or higher were extrapolated by using ROC analysis. The impact of pre-sFLC assay evaluation prior to treatment initiation was used in a retrospective analysis in order to estimate its predictive value on PFS in the case of iFLC and/or FLCr below the cut-off values. Indeed, out of 195 patients in the entire cohort at disease relapse those with pre-iFLC value less than 138 mg/mL had an improved PFS (Figure 4A). The same trend was confirmed in patients with pre-FLCr less than 25 (Figure 4B). Statistical significance was also confirmed in third treatment line for both iFLC and FLCr, while in fourth line iFLC alone remained significant (Table 5).

On the other hand, post-sFLC assay at disease relapse following last treatment line was analyzed for the purpose of estimating its use in predicting disease relapse and relapse type. The study cohort of 229 patients with post-sFLC values at disease relapse was evaluated, confirming, once again, the negative trend in terms of PFS following last therapy both in those with post-iFLC value greater than 138 mg/mL (Figure 4C) and with post-sFLC ratio greater than 25 (Figure 4D). Post-sFLC ratio remained significant after third and fourth treatment line (Table 5). Odds ratio confirmed the association between post-sFLC values and relapse type (biochemical versus clinical), thus revealing a hypothetical connection between sFLC values in course of treatment and development of organ damage at disease relapse. However, neither pre- nor post-sFLC values remained statistically significant in multivariate analysis.

In order to evaluate patients with different characteristics in terms of secretory status, aggressive disease and treatment type, different subcohorts were analyzed by using the abovementioned sFLC cut-off values. Pre-sFLC values, involved chains and ratio remained significant in patients independently from treatment type, doublet or triplet, and in continuous treatment, but had no impact in cases of fixed-duration treatment. The lack of impact, in terms of PFS, was confirmed for post-sFLC values in both fixed-duration treatment and triplet regimens, while, in the rest of the treatment subcohorts, PFS was prolonged in cases of iFLC inferior to 138 mg/mL and with a ratio less than 25 (Table 6). A possible explanation could be that, in cases of fixed-duration treatments, the drug-free period after treatment completion could have an influence in terms of PFS, independently from sFLC alterations. As for the triplet regimen, the combined effect of three drugs could have an effect on slower post-sFLC increase, given that, due to an outpatient treatment in day hospital setting, this subset of patients is monitored more regularly as compared with the doublet regimens, wherein oral drugs prevailed greatly. Similarly, patients with oligo-secretory/micromolecular secretory status tended to be monitored with sFLC on a regular basis, and therefore, the impact of serum free light chain values is not as impactful in the post-sFLC setting as in the pre-sFLC one (Table 6). Finally, given the extremely limited subpopulation with EMD, no correlation with sFLC values was found.

The study was retrospectively designed, without the availability of serial sFLC assay monitoring in the course of treatment for all patients. Furthermore, limitations, such as an extremely heterogeneous population with disease diagnosis in the last 20 years and different treatment regimens throughout treatment lines, make it difficult to strongly confirm the supposed prognostic impact of sFLC assay. Finally, the abovementioned cut-off values, which were obtained by using an N-Latex Siemens assay, cannot be introduced in every-day practice due to the lack of interchangeability between different sFLC methods, especially in case of involved sFLC.

## 5. Conclusions

With the growing number of treatment lines and innovative drugs, the dosage of monoclonal component alone could underestimate disease evolution and could be improved by sFLC assay monitoring. In this work we have evidenced that in more than 20% of normo-secretory patients the secretory status switched to oligo-secretory after the first two lines of treatment. We can, therefore, suggest close periodical monitoring of sFLC to evaluate response both prior to and in the course of treatment and search for clinical disease relapse with organ damage.

## Figures and Tables

**Figure 1 cancers-13-06017-f001:**
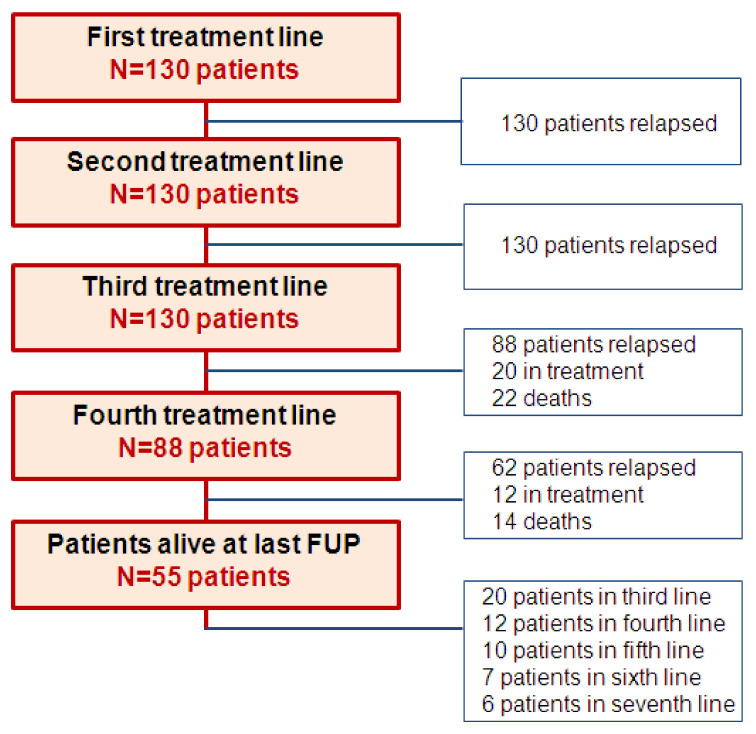
Patient flow diagram from first to fourth treatment line. Abbreviations: FUP—follow-up.

**Figure 2 cancers-13-06017-f002:**
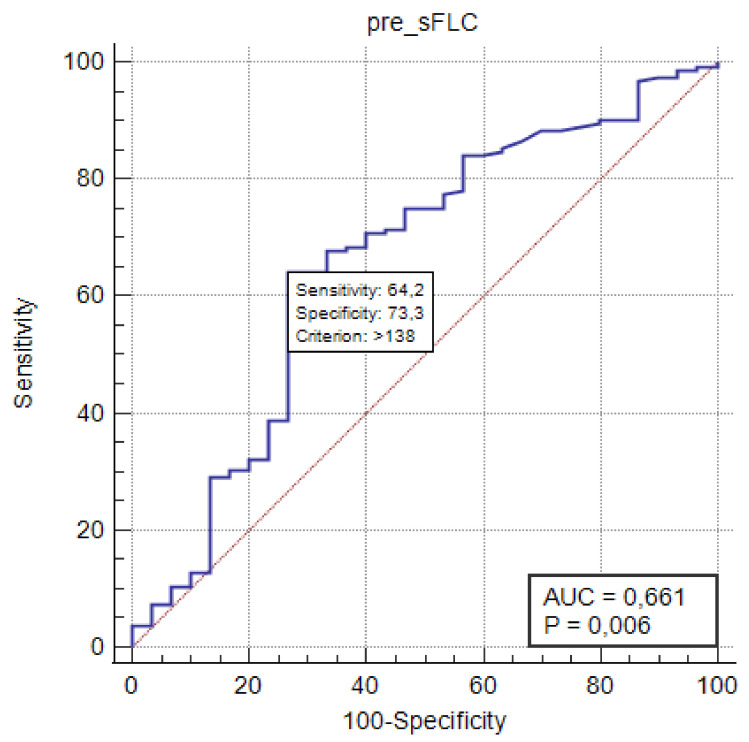
ROC curve analysis of involved pre-sFLC values based on PFS status of the entire cohort throughout four treatment lines. Abbreviations: AUC—area under curve; pre-sFLC—involved serum free light chains prior to treatment start.

**Figure 3 cancers-13-06017-f003:**
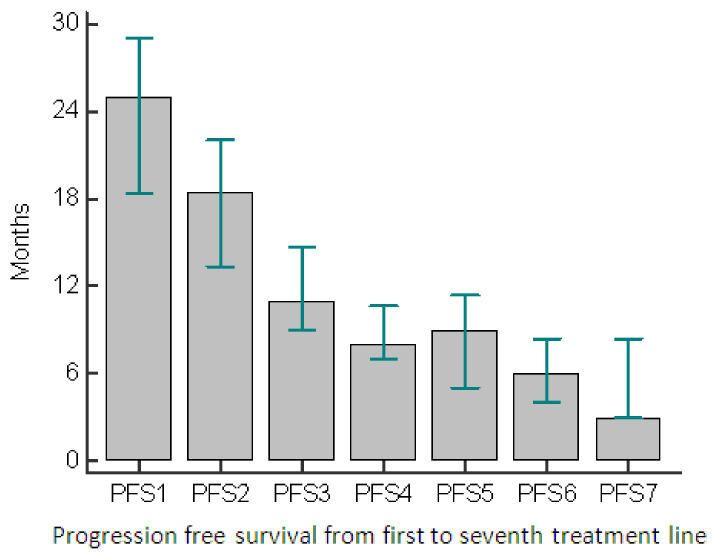
Progression-free survival throughout different treatment lines. Abbreviations: PFS—progression-free survival.

**Figure 4 cancers-13-06017-f004:**
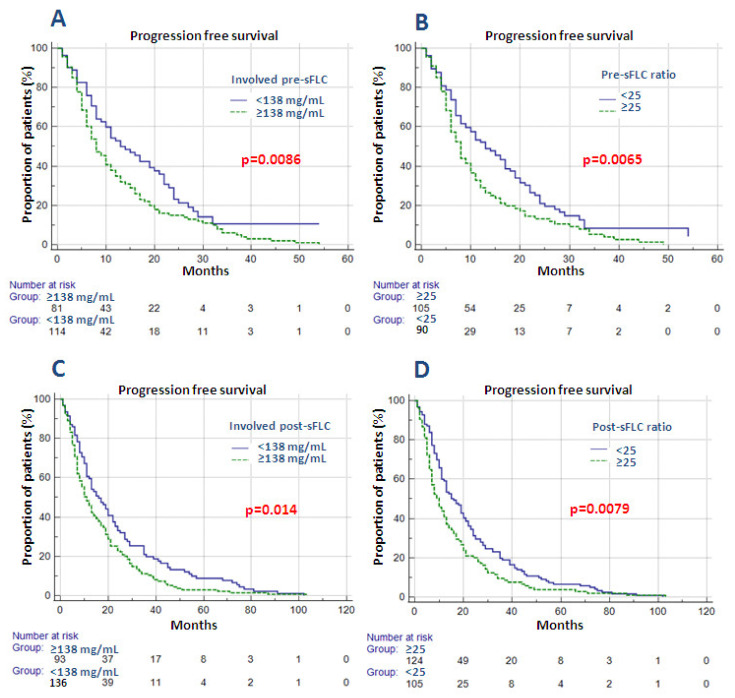
(**A**–**D**) Impact of pre- and post-involved serum free light chains (sFLC) and sFLC ratios on the study cohort in terms of PFS. Abbreviations: Involved pre-sFLC—involved serum free light chains prior to treatment start; Pre-sFLC ratio—serum free light chains ratio prior to treatment start; Involved post-sFLC—involved serum free light chains at disease relapse following last therapy; Post-sFLC ratio—serum free light chains ratio at disease relapse following last therapy.

**Table 1 cancers-13-06017-t001:** Clinical characteristics in 130 multiple myeloma patients at diagnosis.

Age	
median in years (range)	64 (31–80)
<65 years, N (%)	74 (57)
65–75 years, N (%)	47 (36)
>75 years, N (%)	9 (7)
Gender	
male, N (%)	73 (56)
female, N (%)	57 (44)
Time of diagnosis	
years 2000–2005, N (%)	10 (8)
years 2006–2010, N (%)	30 (23)
years 2011–2015, N (%)	63 (48)
years 2016–2019, N (%)	27 (21)
CRAB criteria (130 patients)	
anemia (Hb < 10 g/dL)	64 (49)
bone lesions (>1 lesion)	120 (92)
renal insufficiency (Creatinine > 2 mg/dL)	10 (8)
hypercalcemia (>11.5 mg/dL)	10 (8)
LDH (99 patients)	
normal	84 (85)
elevated	15 (15)
FISH risk stratification (57 patients)	
standard	46 (81)
high	11 (19)
R-ISS staging (60 patients)	
stage I-II	48 (80)
stage III	12 (20)
Extramedullary disease (130 patients)	
no	126 (97)
yes	4 (3)

Abbreviations: LDH—lactate dehydrogenase; FISH—fluorescent in-situ hybridization; R-ISS—revised international staging system.

**Table 2 cancers-13-06017-t002:** Treatment characteristics and secretory status of the study cohort.

Treatment Lines	
median, N (range)	4 (3–8)
≤4 lines, N (%)	72 (55)
>4 lines, N (%)	58 (45)
Stem cell transplantation (63 patients)	
single ASCT, N (%)	32 (51)
double ASCT, N (%)	21 (33)
ASCT + Allogeneic, N (%)	10 (16)
Paraprotein (isotype)	
secreting, N (%)	130 (100)
IgG-heavy chain, N (%)	97 (75)
IgA-heavy chain, N (%)	18 (14)
Secretion type (130 patients)	
normo-secretory, N (%)	106 (82)
oligo-secretory, N (%)	9 (7)
micromolecolar, N (%)	15 (11)
Normo-secretory to unmeasurable switch (106 patients)	
no switch	77 (72)
oligo-secretory switch after ≤3 lines	17 (16)
oligo-secretory switch after >3 lines	4 (4)
micromolecular secretory switch	4 (4)
nonproducer secretory switch	4 (4)

Abbreviations: ASCT—autologous stem-cell transplantation.

**Table 3 cancers-13-06017-t003:** Disease and treatment characteristics from first to fourth treatment line.

		First Line		Second Line		Third Line		Fourth Line	
*Patients N. (%)*		130 (100)		130 (100)		130 (100)		88 (68)	
*Secretory disease*	normo-secretory	130	106 (82)	130	87 (67)	130	82 (63)	88	57 (65)
	oligo-secretory/ micromolecular		24 (18)		43 (33)		49 (37)		31 (35)
*Cytogenetic risk*	standard	55	46 (84)	12	10 (83)	15	10 (67)	8	5 (62)
	high		11 (16)		2 (17)		5 (33)		3 (38)
*Baseline LDH*	normal	99	84 (85)	114	91 (80)	113	92 (81)	78	67 (86)
	increased		15 (15)		23 (20)		21 (19)		11 (14)
*ECOG*	<3	130	104	130	103	130	101	88	53
	≥3		26		27		29		35
*Treatement type*	continuous	130	48 (37)	130	72 (55)	130	91 (70)	88	58 (66)
	fixed duration		82 (63)		58 (45)		39 (30)		30 (34)
*Treatement type*	triplets	50	23 (46)	89	26 (29)	99	23 (23)	58	19 (33)
	doublets		27 (54)		63 (71)		76 (77)		39 (67)
*Response*	CR	130	31 (24)	130	22 (17)	130	14 (11)	88	6 (7)
	VGPR		36 (28)		15 (12)		12 (9)		10 (11)
	PR		42 (32)		61 (47)		55 (42)		32 (36)
	<PR		21 (16)		32 (24)		49 (38)		40 (46)
*Relapse type*	clinical	130	90 (69)	130	96 (74)	109	88 (81)	74	56 (76)
	biochemical		40 (31)		34 (26)		21 (19)		18 (24)
*Extramedullary disease*	no	130	126 (97)	130	126 (97)	130	126 (97)	88	86 (98)
	yes		4 (3)		4 (3)		4 (3)		2 (2)
*Lenalidomide refcractory*	no	130	N.E.	130	111 (85)	130	54 (42)	88	23 (26)
	yes		N.E.		19 (15)		76 (58)		65 (74)

Abbreviations: N.E.—not evaluable; CR- complete response; VGPR—very good partial response; PR—partial response.

**Table 4 cancers-13-06017-t004:** Serum free light chain response evaluation and relapse anticipation from treatment line one to four and of the study cohort.

*Therapy*		First Line		Second Line		Third Line		Fourth Line		Cohort	
*Involved pre-FLC*	<138 mg/mL	28	9 (32)	41	14 (34)	71	33 (46)	55	25 (45)	195	82 (42)
	≥138 mg/mL		19 (68)		27 (66)		38 (54)		30 (55)		113 (58)
*Pre-FLCr*	<25	28	10 (36)	41	23 (56)	71	39 (55)	55	33 (60)	195	105 (54)
	≥25		18 (64)		18 (44)		32 (45)		22 (40)		90 (46)
*Involved post-FLC*	<138 mg/mL	41	14 (34)	71	33 (46)	68	26 (38)	49	19 (39)	229	93 (41)
	≥138 mg/mL		27 (66)		38 (54)		42 (62)		30 (61)		136 (59)
*Post-FLCr*	<25	41	23 (56)	71	39 (55)	68	36 (56)	49	26 (53)	229	124 (55)
	≥25		18 (44)		32 (45)		32 (44)		23 (47)		105 (45)
*First sFLC alteration*	<6 months	12	9 (75)	18	11 (61)	26	16 (62)	24	19 (79)	80	55 (69)
	6–12 months		2 (17)		7 (39)		9 (35)		4 (17)		22 (27)
	>12 months		1 (8)		0		1 (3)		1 (4)		3 (4)
*Anticipation sFLC type*	No alteration	17	5 (29)	35	17 (48)	34	8 (24)	29	5 (17)	115	35 (30)
	sFLC + M-Pr.		8 (47)		7 (20)		17 (50)		13 (45)		45 (40)
	sFLC alone		4 (24)		11 (32)		9 (26)		11 (38)		35 (30)

Abbreviations: Involved pre-FLC—involved serum free light chains prior to treatment start; Pre-FLCr—serum free light chains ratio prior to treatment start; Involved post-FLC—involved serum free light chains at disease relapse following last therapy; Post-FLCr—serum free light chains ratio at disease relapse following last therapy; sFLC – serum free light chains; M-Pr.—monoclonal protein.

**Table 5 cancers-13-06017-t005:** Univariate analysis of progression-free survival (PFS) from 1st to 4th treatment line and of the study cohort, based on serum free light chain values.

Category		*n.*	*Median PFS1, Months (95% CI)*	*p*	*n.*	*Median PFS2, Months (95% CI)*	*p*	*n.*	*Median PFS3, Months (95% CI)*	*p*	*n.*	*Median PFS4, Months (95% CI)*	*p*	*n.*	*Cohort’s Median PFS, Months (95% CI)*	*p*
*Involved pre–FLC*	<138 mg/mL	9	21 (2–28)	0.59	14	11 (6–19)	0.34	33	20 (7–23)	0.0002	25	10 (7–19)	0.03	82	13 (10–20)	0.0086
	≥138 mg/mL	19	16 (8–29)		27	10 (8–20)		38	6 (5–8)		30	7 (5–12)		113	8 (7–10)	
*Pre–FLCr*	<25	10	21 (2–28)	0.98	23	14 (8–19)	0.71	39	17 (7–23)	0.0009	33	9 (7–15)	0.33	105	13 (9–17)	0.0065
	≥25	18	14 (6–26)		18	8 (4–19)		32	6 (5–8)		22	7 (5–12)		90	8 (6–10)	
*Involved post–FLC*	<138 mg/mL	14	28 (13–45)	0.46	33	20 (10–29)	0.16	26	15 (6–22)	0.059	19	11 (6–15)	0.13	93	16 (11–22)	0.014
	≥138 mg/mL	27	24 (11–29)		38	15 (10–24)		42	7 (6–10)		30	7 (5–9)		136	11 (8–14)	
*Post–FLCr*	<25	23	28 (21–35)	0.8	39	20 (11–29)	0.13	36	11 (7–20)	0.03	26	11 (7–15)	0.01	124	16 (12–20)	0.0079
	≥25	18	18 (8–27)		32	14 (9–24)		32	6 (5–10)		23	7 (5–8)		105	11 (7–13)	

Abbreviations: *n*.—number of patients; CI—confidence interval; *p*—*p* value; PFS—progression free survival; Involved pre-FLC—involved serum free light chains prior to treatment start; Pre-FLCr—serum free light chains ratio prior to treatment start; Involved post-FLC—involved serum free light chains at disease relapse following last therapy; Post-FLCr—serum free light chains ratio at disease relapse following last therapy.

**Table 6 cancers-13-06017-t006:** Univariate analysis of the cohort’s PFS divided into different subgroups.

*Category*		*n.*	*Median PFS Triplets, Months (95% CI)*	*p*	*n.*	*Median PFS Doublets, Months 95% CI)*	*p*	*n.*	*Median PFS Continuous, Months (95% CI)*	*p*	*n.*	*Median PFS Fixed- Duration, Months (95% CI)*	*p*	*n.*	*Median PFS Oligo-sec., Months (95% CI)*	*p*
*Involved pre-FLC*	<138 mg/mL	30	24 (8-29)	0.03	34	15 (7-22)	0.02	59	17 (9-24)	0.0002	22	11 (6-16)	0.29	36	20 (10-25)	0.03
	≥138 mg/mL	35	8 (6-10)		57	8 (6-12)		79	7 (6-9)		35	11 (8-17)		50	9 (7-13)	
*Pre-FLCr*	<25	35	18 (11-29)	0.004	48	15 (7-19)	0.02	75	16 (9-20)	0.0001	31	11 (6-16)	0.23	46	18 (10-25)	0.009
	≥25	30	7 (5-9)		43	8 (6-12)		63	7 (5-9)		26	10 (6-16)		40	8 (6-11)	
*Involved post-FLC*	<138 mg/mL	26	10 (8-17)	0.6	42	20 (13-25)	0.001	62	17 (11-22)	0.01	31	16 (11-23)	0.13	35	13 (8-20)	0.2
	≥138 mg/mL	33	9 (6-18)		60	10 (7-13)		80	9 (7-12)		57	14 (10-20)		54	10 (7-17)	
*Post- FLCr*	<25	30	10 (8-16)	0.76	56	19 (13-22)	0.005	80	15 (10-20)	0.007	45	19 (11-23)	0.22	45	12 (8-20)	0.3
	≥25	29	7 (6-16)		46	8 (6-13)		62	8 (6-12)		43	13 (7-18)		44	10 (7-13ù7)	

Abbreviations: *n*.—number of patients; CI—confidence interval; *p*—*p* value; PFS—progression-free survival; Involved pre-FLC—involved serum free light chains prior to treatment start; Pre-FLCr—serum free light chains ratio prior to treatment start; Involved post-FLC—involved serum free light chains at disease relapse following last therapy; Post-FLCr—serum free light chains ratio at disease relapse following last therapy; oligo-sec.—oligo-secretory/micromolecular.

## Data Availability

The data that support the findings of this study are available from the corresponding author, U.M., upon reasonable request.

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
