# Peer review of "Role of Serum Free Light Chain Assay in Relapsed/Refractory Multiple Myeloma. A Real-Life Unicentric Retrospective Study"

_cancers, 2021, doi:10.3390/cancers13236017_

Round 1
Reviewer 1 Report
Brief summary
The current article by Dr Markovic et al role of serum free light chain assay in RRMM. The authors confirm macrofocal relapse or oligosecrtory/ non secretory status which increases with subsequent relapses. They show how a readily able laboratory test, free light chain assay could be used in as a precursor to future clinical relapse.
The study is retrospective with a small size (N=130) and treated heterogeneously. The manuscript is well written.
Major Comments
1.The difference between non secretory and non-producer is the absence of K/L chains by immunohistochemistry. In both Immunofixation is negative. The first paragraph on page 2 in the introduction section seems a little confusing
- Please consider defining micromolecular clearly. https://doi.org/10.1002/ajh.25755. This article shows in a large sample size the macrofocal relapse or relapse with relative bone marrow sparing.
- R ISS stage is determined at diagnosis and remains the same through the disease course. LDH is useful as a component of R ISS staging at diagnosis after that its elevated in relapsed setting including aggressive forms such as secondary plasma cell leukemia. (Page 3 under disease characteristics at diagnosis and relapse)
- What are the results of multivariate analysis? Does a pre-sFLC greater than 138 269 mg/mL and its ratio >25 independently associated with inferior PFS? Are patients treated if these parameters were met?
Minor comments
- What percentage of patients switch to non-secretory MM? What kind of novel imaging techniques are used routinely in your institution
Author Response
Dear reviewer, we thank You for Your valuable comments.
Attached You can find a point by point response.

Reviewer 2 Report
In their manuscript, Markovic et al. retrospectively analyzed the occurrence of oligosecretory/micromolecular escape at disease relapses and its impact on disease outcome. They also evaluated the prognostic role of serum free light chain assay (sFLC) in relapsed/refractory multiple myeloma (RRMM). They first showed disease characteristics such as gender, age, and time of diagnosis, and treatment response and relapse types of 130 RRMM patients analyzed. Next, using ROC analysis, they showed the prognostic significance of pre-involved sFLC (iFLC) greater than 138 mg/mL and ratio <25. Most importantly, they showed that patients with sFLC>138 mg/mL and the FLCr>25 at disease relapse had a higher probability of having clinical relapse. The novelty of this study is that they demonstrated the impact of involved and sFLC ratio on disease relapse. Overall, this manuscript is well written. Only a few minor points need to be improved.
- In the abstract, the authors didn’t clearly define the meaning of FLCr and iFLC, which may cause confusion. Similarly, no description of RRMM is made when the word first appears in the Method section. Please describe these terms accordingly.
- The meaning of involved/uninvolved FLC and the “ratio” is not well explained. Please describe these terms in the introduction and explain why they are important for the evaluation of the disease.
- In Figure 3, the authors showed progression free survival throughout different treatment lines. The title/labeling of Y-axis is missing. Please include this information in Figure3.
Author Response
Dear reviewer, we thank You for Your valuable comments.
Attached You can find our point by point response.

Reviewer 3 Report
The manuscript by Markovic U. et al. present a 20 year period retrospective study on the role of sFLCi and sFLCr as prognostic biomarker of relapse. The tematic is important and the conclusions relevant. However its i pity the cutt-off values may not be used as refence as the authors did not use the monoclonal based assessment recommended by IMWG (binding Site assessemnt) and used the polyclonal based FCL assessment.
Several changes(improvements are required in this manuscript:
1 - In the Abstract - line 27-28 the reference to the ROC curve cutt-off must be clarified as this value will not be recommended for every day practice.
2 - Yet in the Abstract the conslusions taken are too strong for the retrospective nature of the data presented . This need to be said in a more neutral way. based on the data presented.
3- The word "Continuous" treatment should be used and not "Continuative"
4 - Regarding the type of treatments used, it would be important to see the performamce of sFCLi and ratio on the cohort divided by triplet or duplets treatments.
5 - On table 1 (desciption of the Cohort), Myeloma-defining events should be analysed and reported.
6 - On table 2 and/or in the text, the median and intervals of the values of M protein measured in the oligosecretory cohort should be given.
7 - On table 3, CR rates should be shown independently of VGPR.
8 - On line 244, the number of patients should be stated.
9 - The ROC curve should be built also for the oligosecretory cohort. The curve should or not be presented but this information should be referred and commented.
10 - In fig 3. the Y axis name and unit should be detailed.
11- In table 5, the number 5 is missing on "PFS" (column 16).
12 - In Figure 4 , the legend should be extended and it should contain the relevant information to understand the charts without searching in the text.
13 - The authots discuss Extramedullary disease in the Discussion. However any analysis is done or analysed in this sub group. It should be included in table 1 and analysed as a different sub cohort, if possible.
Author Response

(The authors gave the same response as above.)

Round 2
Reviewer 1 Report
The authors have addressed all the concerns.